# Model to Predict Oral Frailty Based on a Questionnaire: A Cross-Sectional Study

**DOI:** 10.3390/ijerph192013244

**Published:** 2022-10-14

**Authors:** Tatsuo Yamamoto, Tomoki Tanaka, Hirohiko Hirano, Yuki Mochida, Katsuya Iijima

**Affiliations:** 1Department of Dental Sociology, Kanagawa Dental University, Yokosuka 238-8580, Japan; 2Institute of Gerontology, The University of Tokyo, Bunkyo-ku, Tokyo 113-8656, Japan; 3Tokyo Metropolitan Geriatric Hospital and Institute of Gerontology, Itabashi-ku, Tokyo 173-0015, Japan; 4Institute for Future Initiatives, The University of Tokyo, Bunkyo-ku, Tokyo 113-0033, Japan

**Keywords:** oral frailty, prediction model, questionnaire, older people, receiver operating characteristic curve, cross-sectional study

## Abstract

A statistical model to predict oral frailty based on information obtained from questionnaires might help to estimate its prevalence and clarify its determinants. In this study, we aimed to develop and validate a predictive model to assess oral frailty thorough a secondary data analysis of a previous cross-sectional study on oral frailty conducted on 843 patients aged ≥ 65 years. The data were split into training and testing sets (a 70/30 split) using random sampling. The training set was used to develop a multivariate stepwise logistic regression model. The model was evaluated on the testing set and its performance was assessed using a receiver operating characteristic (ROC) curve. The final model in the training set consisted of age, number of teeth present, difficulty eating tough foods compared with six months ago, and recent history of choking on tea or soup. The model showed good accuracy in the testing set, with an area of 0.860 (95% confidence interval: 0.806–0.915) under the ROC curve. These results suggested that the prediction model was useful in estimating the prevalence of oral frailty and identifying the associated factors.

## 1. Introduction

Frailty, the most common and troublesome syndrome of an aging population, is a state of vulnerability to a poor resolution of homeostasis after a stressor [1]. It is a consequence of the cumulative decline of several physiological systems during a lifetime [1]. Frailty often progresses, resulting in a higher risk of a disability, falls, fragility fractures, hospital admissions, and early death. Various factors contribute to the development of frailty, including a lack of physical activity [2] and a poor nutritional status [3].

A decline in oral function is highly prevalent among older adults [4], which results in a poor nutritional status [5,6] and is strongly associated with frailty [7,8,9]. A systematic review identified a significant longitudinal association between oral function, including functional dentition with occluding pairs and maximum bite force, and frailty in older people [10].

To increase awareness regarding the importance of oral function in the Japanese population, the concept of oral frailty was introduced in Japan in 2013 [11]. Oral frailty was defined by the Japan Dental Association as a series of phenomena and processes that lead to changes in various oral conditions (number of teeth, oral hygiene, and oral functions) associated with aging, accompanied by a decreased interest in oral health, reduced physical and mental reserve capacities, and an increase in oral frailty leading to an eating dysfunction; the overall effect is a deterioration in physical and mental functions [11].

A few studies have evaluated the prevalence of oral frailty, but these have focused on relatively small populations [12,13]. The assessment of several components of oral frailty such as tongue pressure and articulatory oral motor skills require special equipment; hence, it is difficult to conduct large-scale surveys to clarify regional differences in the prevalence of oral frailty.

Self-reports are suitable for collecting health-related information through large-scale surveys and are useful for planning disease-prevention methods and building healthcare systems [14]. Although self-reports have a few limitations regarding validity [15], self-reports of dental oral health status have been reported to have a degree of validity [16]. Previous studies have shown an association between self-reported oral function and frailty [17,18].

There is little information on the evaluation of oral frailty from self-reported questionnaires. The Oral Frailty Index-8 (OFI-8), an eight-item questionnaire, was developed to identify older adults at risk of oral frailty [19]. However, the questions used in this disquisition include items that were not used in several previously conducted surveys [20].

The Japan Gerontological Evaluation Study was one of the largest nationwide gerontological questionnaire surveys aimed at understanding the overall status of the older population and how this population had changed over time [20]. The study included questions related to oral health such as “number of teeth”, “difficulty eating tough foods compared to six months ago”, and “recent history of choking on tea or soup”, which are components of oral frailty [21]. If oral frailty could be determined by these questions in addition to age and sex, a large survey could be conducted to examine the regional differences in oral frailty and the associated factors. Therefore, this study aimed to develop and test the validity of a formula to determine oral frailty using these questions.

## 2. Materials and Methods

### 2.1. Study Population

The present cross-sectional study was conducted using the baseline data of a screening program for oral frailty [22]. The participants were older people who visited 25 dental clinics in Kanagawa Prefecture, Japan, and participated in a screening program conducted between August 2018 and January 2019 to assess the eligibility of an oral frailty measures program in Kanagawa Prefecture.

The study included individuals ≥ 65 years of age. The exclusion criterion was missing data on items representing oral frailty. Of the 848 people who participated in the screening, one participant aged < 65 years and four with missing data on difficulties in chewing tough foods, tongue pressure, and gum test scores were excluded. A total of 843 people aged ≥ 65 years (mean age: 77.9 years old; standard deviation: 5.4 years old; 385 males and 458 females) were included in the study.

### 2.2. Definition and Assessment of Oral Frailty

The presence or absence of oral frailty was evaluated based on six items [22], which was a modification of the method described by Tanaka et al. [21]. Oral frailty was considered to be present if at least three of the following six criteria were met: (1) <20 teeth; (2) chewing ability score of <4; (3) articulation of the sound “ta” at less than six times per second; (4) tongue pressure of <30.0 kPa; (5) subjective difficulty in eating tough foods; and (6) subjective difficulty in swallowing. The examination methods and criteria of the six items from (1) to (6) were as follows:

(1) The number of teeth present was evaluated by 25 dentists. Fewer than 20 teeth was considered to be one item of oral frailty.

(2) A chewing ability score was obtained using a gum test, in which the patients were instructed to chew color-changeable chewing gum (Lotte Confectionery, Seoul, Korea) for one minute. The chewing function was assessed based on the changes in the gum color using a five-level (from score 1 (worst) to score 5 (best)) color chart [22,23]. A chewing ability score of <4 was considered to be an item of oral frailty.

(3) The repetitive articulatory rate—i.e., oral diadochokinesis (ODK)—was measured using an oral function measuring device (Kenko-kun handy, Takei Scientific Instruments Co., Ltd., Niigata, Japan). This aided the evaluation of the articulatory oral motor skill. In the ODK test, participants were instructed to repeatedly and rapidly pronounce “ta” monosyllables for five seconds. The average number of repetitions per second was calculated [22]. The articulation of the sound “ta” less than six times per second was considered to be one item of oral frailty.

(4) The force produced by the contact between the anterior part of the hard palate and tongue, referred to as tongue pressure, was measured using a JMS tongue pressure meter (TPM-01, JMS Co., Ltd., Hiroshima, Japan) [22]. A balloon probe was automatically adjusted to a predetermined pressure and placed over the tongue. Thereafter, the participants were instructed to push the tip of the tongue upward against the palate for approximately seven seconds at a maximum force. The maximum pressure was recorded. A tongue pressure of <30.0 kPa was considered to be an item of oral frailty.

(5 and 6) Questions from the Kihon checklist were used to determine a subjective difficulty in eating tough foods and swallowing [24]. The questions “Do you have any difficulties eating tough foods in compared with six months ago?” and “Have you choked on your tea or soup recently?” were asked to assess the self-perceived oral function. A “yes” answer for each question was considered to be an item of oral frailty.

### 2.3. Statistical Analysis

The data were split into a training and a testing set (a 70/30 split) using random sampling. The training set was used to develop a logistic regression model. The model was evaluated using a test set. In the descriptive statistics, the characteristics associated with oral frailty were analyzed in both the training and testing datasets to ensure homogeneity. In the training set, the association of oral frailty with age, sex, and the characteristics of oral frailty was examined using a univariate logistic regression analysis. A prediction model was developed using a stepwise logistic regression of the training set. In the model, we used variables that could be obtained from the questionnaire such as age, sex, number of teeth present, difficulty in eating tough foods compared with six months ago, recent history of choking on tea or soup, and a dry mouth. The statistical significance was set at a *p*-value < 0.05. Using a logistic regression equation, a formula for calculating the probability of oral frailty was developed.

The discrimination of the model was evaluated using the area under the curve (AUC) in a receiver operating characteristic (ROC) analysis. In addition, the prediction performance of the model was evaluated using sensitivity, specificity, the positive predictive value (PPV), and the negative predictive value (NPV) in both the training and testing sets.

IBM SPSS Modeler (version 18.3, SPSS Japan Inc., Tokyo, Japan) was used to prepare the training and testing sets. Other statistical analyses were performed using IBM SPSS Statistics (version 27, SPSS Japan Inc.) with a significance level of 5%.

### 2.4. Ethical Approval

All data used in the analysis were anonymous and the requirement for informed consent was waived based on the Ethical Guidelines for Medical and Biological Research Involving Human Subjects in Japan. The corresponding author signed a memorandum of understanding with Kanagawa Prefecture regarding the use of the screening data. The ethics committee of Kanagawa Dental University issued ethical clearance for a secondary analysis of the screening data (approval No. 856).

## 3. Results

### 3.1. Characteristics of the Participants

The demographic and clinical characteristics of the participants in the training and testing sets are shown in Table 1. No statistically significant differences were observed between the two datasets.

### 3.2. Development of the Prediction Model

Based on the univariate analysis, an age ≥ 85 years (odds ratio (OR) = 3.94; 95% confidence interval (CI): 1.97–7.90; reference 65–74 years old), the presence of <20 teeth (OR = 9.42; 95% CI: 6.15–14.42), an oral diadochokinesis of <6.0 times/s (OR = 6.32; 95% CI: 4.18–9.56), a tongue pressure of <30.0 kPa (OR = 6.95; 95% CI: 4.49–10.75), a gum test score of <4 (OR = 9.79; 95% CI: 6.37–15.05), difficulty eating tough foods compared with six months ago (OR = 8.48; 95% CI: 5.35–13.44), a recent history of choking on tea or soup (OR = 5.98; 95% CI: 3.85–9.28), and a dry mouth (OR = 1.53; 95% CI: 1.03–2.29) significantly increased the OR for oral frailty (Table 2).

The first and final models obtained using a multivariable logistic regression analysis with corresponding adjusted OR and 95% CI in the training set are shown in Table 3. In the final model, an age ≥85 years (OR = 5.28; 95% CI: 2.06–13.57), the presence of <20 teeth (OR = 12.97; 95% CI: 7.41–22.70), difficulty eating tough foods compared with six months ago (OR = 7.58; 95% CI: 4.19–13.70), and a recent history of choking on tea or soup (OR = 11.74; 95% CI: 6.34–21.75) significantly increased the probability of oral frailty.

Based on the logistic regression equation, a multivariate logistic regression predictive model was developed to determine the risk of oral frailty.

*p* = (EXP (0.477 × (aged 75–84 years: yes: 1, no: 0) + 1.665 × (aged ≥ 85: yes: 1, no: 0) + 2.563 × (presence of <20 teeth: yes: 1, no: 0) + 2.025 × (difficulty eating tough foods compared with six months ago: yes: 1, no: 0) + 2.463 × (recent history of choking on tea or soup: yes: 1, no: 0) − 3.983))/(1 + EXP (0.477 × (aged 75–84 years: yes: 1, no: 0) + 1.665 × (aged ≥ 85: yes: 1, no: 0) + 2.563 × (presence of <20 teeth: yes: 1, no: 0) + 2.025 × (difficulty eating tough foods compared with six months ago: yes: 1, no: 0) + 2.463 × (recent history of choking on tea or soup: yes: 1, no: 0) − 3.983)).

### 3.3. Screening Performance of the Prediction Model

The ROC curve of the oral frailty prediction model for the testing set is shown in Figure 1.

The screening performance characteristics of the model in the training and testing sets are presented in Table 4. The AUC of the final model for the training and testing sets was 0.890 and 0.860, respectively. The optimal threshold cutoff value was 0.1824 for both sets, which was determined by the highest Youden index value. The sensitivities of the training and testing sets were 0.94 and 0.90, respectively. The specificities of the training and testing sets were 0.67 and 0.66, respectively. The accuracies of the training and testing sets were 0.74 and 0.73, respectively.

## 4. Discussion

We developed a prediction model of oral frailty based on information collected from the Japan Gerontological Evaluation Study. The model, which included variables such as age, the number of teeth present, difficulty eating tough foods compared with six months ago, and a recent history of choking on tea or soup, showed an AUC of 0.860 in the testing set with a high accuracy. Moreover, the screening performance of the model for the training and testing sets was similar, demonstrating the validity of the model. Therefore, the prediction model might prove to be useful in estimating the prevalence of oral frailty using large surveys and questionnaires.

The components used to identify participants with oral frailty included questions regarding difficulty eating tough foods compared with six months ago and a recent history of choking on tea or soup. These questions were used in the Kihon checklist, which was introduced by the Japanese Ministry of Health, Labour and Welfare in 2006 to identify vulnerable older adults who were at a higher risk of becoming dependent. The assessment was performed so that adequate measures could be taken to prevent frailty and disability in these individuals [24]. Most municipalities already have experience collecting information using these questions. In addition, the Kihon checklist has also been used as one of the tools to determine individuals with frailty [25]. It might be easy to estimate the prevalence of oral frailty among individuals by adding questions regarding the number of teeth present and performing the calculation using a prediction model.

In addition, the two questions of difficulty eating tough foods compared with six months ago and a recent history of choking on tea or soup are associated with general health in older people. Difficulty in eating is a predictor of incident depressive symptoms, physical frailty, sarcopenia, and disability [23,26]. Choking is a predictor of incident falls, respiratory diseases, physical frailty, sarcopenia, and disability [21,27,28]. These reported findings suggest the importance of the two aforementioned questions as tools to evaluate oral function, which relates to general health in older individuals.

Two variables, sex and dry mouth, were excluded from the final logistic regression model. This was in agreement with the lack of significant difference in the prevalence of oral frailty between males and females in previous studies [19,29]. In the present study, a dry mouth was significantly associated with oral frailty in the univariate model; however, the degree of significance was weaker compared with the other variables in the final logistic regression model. These findings were in accordance with the Oral Frailty Index-8, which has less weightage for a dry mouth as a variable compared with the variables of having difficulties eating tough foods compared with six months ago and a recent history of choking on tea or soup [19].

Although the prediction model had a high screening performance, the present study had a few limitations. First, the data on the number of teeth present in the current study were based on clinical examinations. As the prediction model was designed to predict oral frailty based on information on the number of teeth present obtained from the questionnaire, the actual accuracy was expected to be low. It could be argued that the questionnaire did not provide a complete and accurate picture of the differences in the number of teeth. However, a self-reported number of teeth is a well-established and reliable measure that has been used in national epidemiological surveys [16]. A high level of agreement was reported between the self-reported and examined number of teeth (Pearson correlation coefficient: *r* = 0.97) in 50 community-dwelling individuals aged ≥ 70 years in the United States [30]. Second, the definition of oral frailty varies between researchers [31]. We used the modified definition [22] proposed by Tanaka et al. [21], which is one of the most popular methods used to evaluate oral frailty. Third, the present study only included patients who visited dental offices. Therefore, the results of the present study cannot be generalized for older adults in Japan. In addition, the development of an algorithm using people with disabilities and people < 65 years of age is required to corroborate this model.

## 5. Conclusions

A validated model was developed to predict oral frailty based on the following variables, which were assessed using a questionnaire: age, number of teeth present, difficulty eating tough foods compared with six months ago, and a recent history of choking on tea or soup. The prediction model might be useful in estimating the prevalence of oral frailty using data obtained from large surveys and questionnaires.

## Figures and Tables

**Figure 1 ijerph-19-13244-f001:**
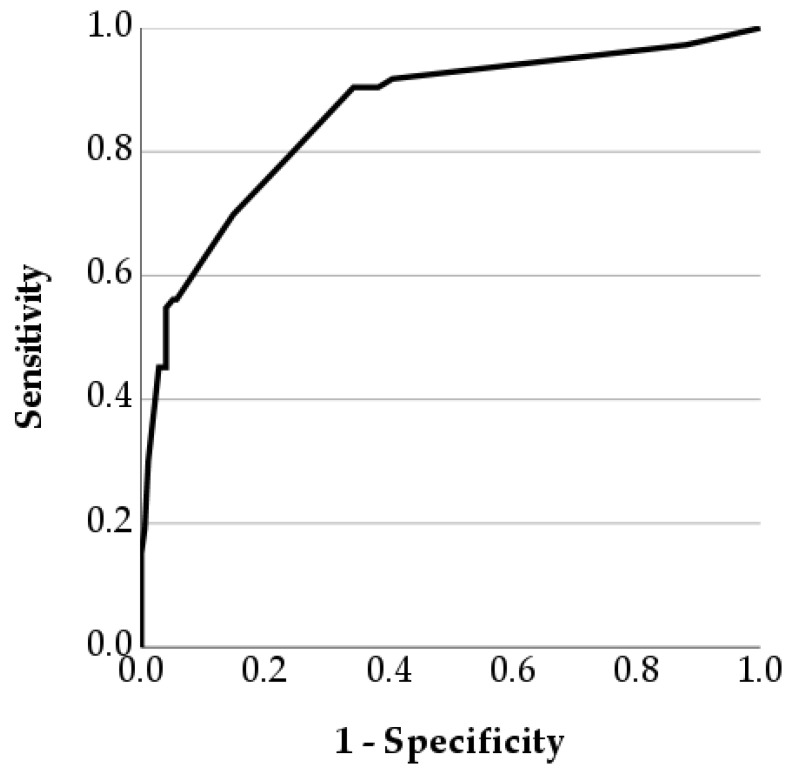
Receiver operating characteristic (ROC) curve of the oral frailty prediction model for the testing set. The area under the curve (95% confidence interval) was 0.860 (0.806–0.915).

**Table 1 ijerph-19-13244-t001:** Demographic and clinical characteristics of subjects in the training and testing sets.

Characteristics		Training Set (*n* = 595)		Testing Set (*n* = 248)		*p*-Value ^3^
		*n*	%	*n*	%	
Age group (years)	65–74	105	17.6	47	19.0	0.492
	75–84	421	70.8	179	72.2	
	≥85	69	11.6	22	8.9	
Sex	Female	330	55.5	128	51.6	0.172
	Male	265	44.5	120	48.4	
Number of teeth present	≥20	407	68.4	166	66.9	
	<20	188	31.6	82	33.1	0.367
Oral frailty	No	453	76.1	175	70.6	
	Yes	142	23.9	73	29.4	0.055
Oral diadochokinesis (times/sec)	≥6.0	375	63.0	153	61.7	
	<6.0	220	37.0	95	38.3	0.386
Tongue pressure (kPa)	≥30.0	340	57.1	144	58.1	
	<30.0	255	42.9	104	41.9	0.433
Gum test score	≥4	445	74.8	177	71.4	
	<4	150	25.2	71	28.6	0.173
Difficulty eating tough foods ^1^	No	489	82.2	204	82.3	
	Yes	106	17.8	44	17.7	0.533
Choking ^2^	No	481	80.8	190	76.6	
	Yes	114	19.2	58	23.4	0.099
Having a dry mouth	No	419	70.4	170	68.5	
	Yes	176	29.6	78	31.5	0.322

^1^ Difficulty eating tough foods compared with six months ago; ^2^ recent history of choking on tea or soup; ^3^ chi-squared test or Fisher’s exact test.

**Table 2 ijerph-19-13244-t002:** Association of age, sex, and each characteristic of oral frailty with oral frailty in the training set.

Variables		Total	With Oral Frailty		OR ^3^	95% CI ^4^		*p*-Value ^5^
		*n* = 595	*n* = 142	%		Lower	Upper	
Age group (years)	65–74	105	18	17.1	1.000			
	75–84	421	93	22.1	1.370	0.785	2.393	0.268
	≥85	69	31	44.9	3.943	1.968	7.898	<0.001
Sex	Female	330	80	24.2	1.000			
	Male	265	62	23.4	0.954	0.653	1.395	0.810
Number of teeth present	≥20	407	43	10.6	1.000			
	<20	188	99	52.7	9.416	6.147	14.424	<0.001
Oral diadochokinesis (times/s)	≥6.0	375	43	11.5	1.000			
	<6.0	220	99	45.0	6.317	4.175	9.558	<0.001
Tongue pressure (kPa)	≥30.0	340	33	9.7	1.000			
	<30.0	255	109	42.7	6.945	4.489	10.746	<0.001
Gum test score	≥4	445	55	12.4	1.000			
	<4	150	87	58.0	9.792	6.370	15.052	<0.001
Difficulty eating tough foods ^1^	No	489	77	15.7	1.000			
	Yes	106	65	61.3	8.483	5.353	13.443	<0.001
Choking ^2^	No	481	80	16.6	1.000			
	Yes	114	62	54.4	5.976	3.850	9.278	<0.001
Having a dry mouth	No	419	90	21.5	1.000			
	Yes	176	52	29.5	1.533	1.029	2.285	0.036

^1^ Difficulty eating tough foods compared with six months ago; ^2^ recent history of choking on tea or soup; ^3^ odds ratio; ^4^ confidence interval; ^5^ chi-squared test or Fisher’s exact test.

**Table 3 ijerph-19-13244-t003:** Results of logistic regression analyses with stepwise variable selection for oral frailty in the training set.

Step	Independent Variables	B ^3^	SE ^4^	OR ^5^	95% CI ^6^		*p*-Value
					Lower	Upper	
Step 1	Age group (years) (reference: 65–74)						
	75–84	0.464	0.372	1.590	0.767	3.296	0.213
	≥85	1.655	0.483	5.234	2.031	13.484	0.001
	Sex (reference: female)						
	Male	0.239	0.263	1.270	0.758	2.127	0.364
	Number of teeth present (reference: ≥20)						
	<20	2.590	0.289	13.323	7.568	23.454	<0.001
	Difficulty eating tough foods ^1^ (reference: no)						
	Yes	2.050	0.306	7.769	4.268	14.139	<0.001
	Choking ^2^ (reference: no)						
	Yes	2.461	0.320	11.721	6.262	21.941	<0.001
	Having a dry mouth (reference: no)						
	Yes	−0.027	0.281	0.973	0.562	1.687	0.923
	Constant	−4.086	0.453	0.017			<0.001
Step 3	Age group (years) (reference: 65–74)						
	75–84	0.477	0.372	1.611	0.777	3.342	0.200
	≥85	1.665	0.481	5.283	2.057	13.567	0.001
	Number of teeth present (reference: ≥20)						
	<20	2.563	0.286	12.968	7.410	22.698	<0.001
	Difficulty eating tough foods ^1^ (reference: no)						
	Yes	2.025	0.302	7.578	4.190	13.704	<0.001
	Choking ^2^ (reference: no)						
	Yes	2.463	0.315	11.738	6.335	21.751	<0.001
	Constant	−3.983	0.431	0.019			<0.001

^1^ Difficulty eating tough foods compared with six months ago; ^2^ recent history of choking on tea or soup; ^3^ coefficients; ^4^ standard errors; ^5^ odds ratios; ^6^ confidence intervals. Step 1: The first model, which included all independent variables. Step 3: The final model.

**Table 4 ijerph-19-13244-t004:** Screening performance characteristics of the model in the training and testing sets.

Characteristics	Training Set	Testing Set
Cut-point	0.182	0.182
Sensitivity	0.937	0.904
Specificity	0.673	0.657
Positive predictive values	0.473	0.524
Negative predictive values	0.971	0.943
Area under the curve	0.890	0.860
Accuracy	0.736	0.730

## Data Availability

Data can be obtained by following the prescribed procedures for Kanagawa Prefecture.

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
