# Peer review of "Model to Predict Oral Frailty Based on a Questionnaire: A Cross-Sectional Study"

_ijerph, 2022, doi:10.3390/ijerph192013244_

Round 1

Reviewer 1 Report

The aim of manuscript titled Model to Predict Oral Frailty Based on Questionnaire: A Cross- 2 

Sectional Study was to develop and validate a predictive model to assess oral frailty.

Generally speaking the topic is interesting, however there some areas and sections in the manuscript that should be improved before it can be evaluated for publication. Especially introduction and deiscussion sections need to be rewritten, and the reference list should be improved.

Introduction

This section gives very brief introduction and focuses mainly on the works very similar in nature to this being reviewed. However, lots of general information is missing, together with scientific background and explanation of rationale. Quite rececntly, a systematic review of longitudinal studies has been published (https://doi.org/10.1111/ger.12406). Please read it and use it as an example to rebuild your manuscript. Cite this article in the referencecs. In this section the rationale for patient subjective evaluation should also be mentioned (doi: 10.1038/sj.ebd.6401323; doi: 10.1111/ger.12513). In this section the rationale for questionnaire studies, their advantages and disadvantages should be discussed briefly (https://doi.org/10.17219/dmp/91774).

Materials and methods

This section is also very brief. Please elaborate on study design, inclusion and exclusion criteria, be more precise with respect to evaluated parameters (give definitions fe. chewing ability scores), including how and when they were assessed. Some readers might not be familiar with the previous work of the Authors and for those such a succint description could be hard to follow.

Results

This section is well-executed.

Discussion

This section needs to be rewritten. This is by and large too short and poorly structured. There is almost no comparison with other studies and the important references to the litearatue are missing. Moreover, the majority of this part is dedicated to limitations of this study. It would be advisable to highlight the strenghts of this research and what it brings to scientifc knowledge regarding this area. The outcomes should be presented in broad light and the reference list should be lengthen and updated (doi: 10.1371/journal.pone.0178383; doi: 10.1111/jgs.15175; doi: 10.1186/s12877-015-0134-9). 

Author Response

Generally speaking, the topic is interesting, however there some areas and sections in the manuscript that should be improved before it can be evaluated for publication. Especially introduction and discussion sections need to be rewritten, and the reference list should be improved.

Response: Thank you for your valuable comment. We have rewritten the Introduction (lines 35-39 and 52-56) and Discussion sections (lines 209-226), and improved the reference list as per your suggestion.

Introduction

This section gives very brief introduction and focuses mainly on the works very similar in nature to this being reviewed. However, lots of general information is missing, together with scientific background and explanation of rationale. Quite recently, a systematic review of longitudinal studies has been published (https://doi.org/10.1111/ger.12406). Please read it and use it as an example to rebuild your manuscript. Cite this article in the references. In this section the rationale for patient subjective evaluation should also be mentioned (doi: 10.1038/sj.ebd.6401323; doi: 10.1111/ger.12513). In this section the rationale for questionnaire studies, their advantages and disadvantages should be discussed briefly (https://doi.org/10.17219/dmp/91774).

Response: Thank you for your comment. We have rewritten the Introduction section ((lines 35-39 and 52-56) and added the suggested article in the reference list, per your recommendation.

Materials and methods

This section is also very brief. Please elaborate on study design, inclusion and exclusion criteria, be more precise with respect to evaluated parameters (give definitions fe. chewing ability scores), including how and when they were assessed. Some readers might not be familiar with the previous work of the Authors and for those such a succinct description could be hard to follow.

Response: Thank you for your valuable comments. We have revised the materials and methods section as per your suggestion, and included the description of parameters, along with the methods of their assessment (lines 89-116). Moreover, the study design (lines 72-73) along with inclusion and exclusion criteria (lines 77-78) have also been added.

Results

This section is well-executed.

Response: Thank you for your encouraging comment.

Discussion

This section needs to be rewritten. This is by and large too short and poorly structured. There is almost no comparison with other studies and the important references to the literature are missing. Moreover, the majority of this part is dedicated to limitations of this study. It would be advisable to highlight the strengths of this research and what it brings to scientific knowledge regarding this area. The outcomes should be presented in broad light and the reference list should be lengthen and updated (doi: 10.1371/journal.pone.0178383; doi: 10.1111/jgs.15175; doi: 10.1186/s12877-015-0134-9).

Response: Thank you for your valuable comment. We have rewritten the Discussion section (lines 209-226) in accordance with your suggestion and updated the reference list. One of the recommended references has been cited in the introduction section. However, incorporating the other suggested references was an arduous task and could not be added in the discussion section. It would require extensive modifications of the manuscript, which could obscure the intended connotation, so we only went over their contents.

Reviewer 2 Report

This manuscript is very interesting and relevant to add to the literature; however, the authors must improve the discussion and conclusion sections without compromising the other sections. If a word count is a challenge, try to be more succinct in the introduction and perhaps a little bit in the Methods, although I would try to leave the Methods section as is as it explains well the process. 

Author Response

This manuscript is very interesting and relevant to add to the literature; however, the authors must improve the discussion and conclusion sections without compromising the other sections. If a word count is a challenge, try to be more succinct in the introduction and perhaps a little bit in the Methods, although I would try to leave the Methods section as is as it explains well the process.

Response: Thank you for your valuable comment. We have revised the Discussion and Conclusion sections (lines 209-226 and 256-257) as per your suggestion. In addition, the Methods section has also been elaborated (lines 89-116).

Round 2

Reviewer 1 Report

I recommend publishing the manuscript since the authors applied all changes suggested in the first round of revision.